# It Is Home: Perceptions, Community, and Narratives about Change

**Michael R. Cope *** , **Haylie M. June, Scott R. Sanders, Greta L. Asay, Hannah Z. Hendricks, Elizabeth Long-Meek and Carol Ward**

Department of Sociology, Brigham Young University, 2008 JFSB, Provo, UT 84602, USA
* Correspondence: michaelrcope@byu.edu

**Abstract:** Utah's Heber Valley has experienced rapid and (relatively) sustained growth since the 1990s, in part due to being chosen as a host venue for the 2002 Winter Olympics. As conditions in the Valley changed by virtue of this growth, individuals had to redefine their relationship with their community at large, as well as what community means to them individually. As individuals integrate new conditions into their imagined communities, they are also required to imagine communities in ways they never have before. The community's story is rewritten simultaneously along with individuals' own stories. These changing stories are shaped and indicated by the reconstruction of residents' narratives about their community, i.e., their community stories. In this paper, we (1) explore how Heber Valley residents' narratives change as a result of preparing for, participating in, and recovering from the Olympics, (2) verify these findings using survey data gathered during the same time period, and (3) examine how changes in residents' narratives in Heber Valley impacted the subjective evaluation of community. To do so, we rely on longitudinal data gathered among principal communities in Heber Valley with additional data generated from a hermeneutic content analysis of archival data found in the area's community newspaper (*The Wasatch Wave*). Survey data were gathered once a year over a five-year period from February 1999 through February 2003, with additional waves gathered in February 2007, 2012, and 2018. Our results indicated that the community narratives did change as a result of the Olympics, our survey data verified these community changes over time, and changes in residents' community stories impacted survey responses when residents were asked about community sentiments.

**Keywords:** community; rural; urbanizing; narrative; community attachment; community satisfaction; community desirability





## 1. Introduction

The concept of "place" has been discussed since Aristotle (Devine-Wright 2009). While place can be tied to a geographical location, it is different from related concepts such as "space" or "environment" because it also considers the social meanings and symbolic interactions residents experience (Tuan 1977; Devine-Wright 2009). Place has also been defined as an environment that "gathers human experiences, meanings, and actions" (Seamon 2018, 2020). Similarly, community is not simply the geographic location in which someone resides, but the social sphere that connects people (Wilkinson 1991; Colling et al. 2017). Where place refers more to the individual level, community is very much a collective effort by many individuals in a shared space.

Community matters. However, "[t]rying to study community is like trying to scoop jello [sic] up with your fingers. You can get hold of some, but there's always more slipping away from you" (Pelly-Effrat 1974, p. 1). Several problems can arise when studying and conceptualizing community, as it is rarely stable enough to be easily identifiable (Puddifoot 1995). The boundaries of a community are defined by the groups that make up that community. Community boundaries do change over time, as the boundaries are often imposed by those who belong to that community based on criteria that can

change. As conditions change, individuals have to redefine their relationship with their community at large, as well as what community means to them individually. They rewrite the community's story as they rewrite their own stories.

In this paper, we offer a case study exploration of Utah's Heber Valley, an area that has experienced rapid and (relatively) sustained growth since the 1990s, in part due to being chosen as the host venue for the 2002 Winter Olympics. Community disruptions of this sort can be viewed as a transition from one perceived community to another. Definitions of place and community are altered to adapt to the event that impacted that area, as that influential event is part of a "new" community in the same geographic location.

After the significant event, the community must be understood in the local context and narratives of the community after said event. The physical location has changed to accommodate the event, and the definitions of community have changed as well. As people harmonize new conditions with their imagined communities, they are also required to imagine communities in ways they never have before. The community's story is rewritten simultaneously with the individual's own stories. These impacts are shaped and indicated by the reconstruction of residents' narratives about their community, i.e., their community stories.

In the present study, we (1) explore how Heber Valley residents' narratives change as a result of preparing for, participating in, and recovering from the Olympics, (2) verify these findings using survey data gathered during the same time period, and (3) examine how changes in residents' narratives in Heber Valley impacted the subjective evaluation of community. To do so, we rely on longitudinal data gathered among the five principal communities of Utah's Heber Valley (Heber, Midway, Daniel, Charleston, and Center) along with data generated from a hermeneutic content analysis of archival data found in the area's community newspaper (*The Wasatch Wave*).

## 2. Background

As humans, we are story-telling animals (MacIntyre 1997), and we tell stories through place; our sense of community is developed through shared narratives that connect us to one another. As Card put it, it is "in large part through shared stories that communities create themselves and bind themselves together" (Card 1990, p. 273). Narratives constitute community because they "explain a group to itself", and help residents understand the community culture, values, and ways of life (Hinchman and Hinchman and Hinchman 1997, p. 235). "Community as story" refers to the community narratives that members of a community create; it is "portrayed through the observed or imagined relations between specific actors who occupy a particular place in time and the inherited stories of that place" (Cope et al. 2019, p. 2). Community narratives are integral to a group's identity and memory (Hinchman and Hinchman and Hinchman 1997) because they provide the members with a story with which to identify and define themselves.

Individuals maintain their personal stories even as they add to and make up the community story: (Flynn 1991). Individual actors take part in the construction and maintenance of the imagined community (Phillips 2002). However, individual stories are not necessarily replications of the community story and may not even seem to 'fit in' with the community story (Hinchman and Hinchman and Hinchman 1997). Individual stories are reconciled with others' stories to create the collective community narrative (Cope et al. 2019). Since we, as individual actors, have the power to reshape and redefine our community narrative, individual stories and community narratives are interconnected. The overarching community narrative actively shapes and is shaped by the individual stories. These narratives socialize people: through narratives, people learn and potentially internalize the accepted culture, practices, norms, and ways of thinking of the group (Hinchman and Hinchman and Hinchman 1997).

In response to internal and external changes in the community, the narratives likewise change. When an event or other condition changes within the community, the perception, meaning how residents view their community and what sentiments they have towards

it, undergoes a transition, becoming, in a sense, a new community in the same location (Greider et al. 1991); the "old" conditions of the community are maintained and incorporated in the community narrative. In rural communities, transitions are often brought about by population growth resulting in socio-cultural changes (Greider et al. 1991). While the conditions of change may be externally imposed, the community members' responses are internally constructed and within their control (Cope et al. 2019). Changes within the community narrative require individuals to redefine their relationship with the community to accommodate the new conditions or events, thereby adjusting and modifying their own story as well as the community story. Hirschman (1970) describes how individuals can cope with changes to their community through three avenues: "exit", where the individual withdraws; "voice", which involves communication to attempt to repair or address the issues; or "loyalty", where the individual accepts the change. Within any community, individuals will likely approach their perception of the community's story using these options.

Community literature has examined social change in rural areas. For example, boomtown literature has provided a framework for what happens in a community when disruption occurs. During the 1970s and 1980s, much research was conducted to investigate the effect of rapid population growth in these boomtown communities (see, e.g., Gold 1974; Gilmore and Duff 1975; Gilmore 1976; Thompson 1979; Freudenburg et al. 1982; England and Albrecht 1984; Krannich and Greider 1984). During this time, researchers found that the development and disruption taking place in these communities had social-psychological effects on the local residents and the feelings they had towards their community (Krannich et al. 1985; Freudenburg 1986; Brown et al. 1989; Krannich et al. 1989; Krannich and Greider 1990). Recent literature on social change in rural areas has found that residents in boomtowns can continue to have a strong sense of community, despite the changes occurring (Ward et al. 2020). However, other case studies have found that social disruption causes a decrease in cohesion in the community and a creation of "insider" and "outsider" categories (O'Connor 2015).

While mega-events, such as the Olympics, are not perfectly comparable to boomtowns, they do provide another example of the effects of social change in rural areas. While research on the social impacts the Olympics have on their host communities is limited, research has found that residents' feelings toward their community are impacted by hosting a mega-event (Kim et al. 2006; Ritchie et al. 2009; Kaplanidou et al. 2013; Cope et al. 2015).

The purpose of the present study is to contribute to the community literature and examine narratives of social change before, during, and after the 2002 Winter Olympics. To do so, we use survey data and textual data gathered from local newspapers to (1) explore how Heber Valley residents' narratives change as a result of preparing for, participating in, and recovering from the Olympics, (2) verify these findings using survey data gathered during the same time period, and (3) examine how changes in residents' narratives in Heber Valley impacted the subjective evaluation of community.

## 3. Study Setting

### 3.1. Historical Settlement

The Heber Valley is located approximately forty miles away from Salt Lake City, sitting in Utah's Wasatch Mountain Range. The area was used by indigenous peoples for gathering food during summer fishing before non-Native peoples settled the area (Bancroft 1883; Mortimer 1963; Embry 1996). The first record of non-Native peoples visiting the area takes place in 1776, per the records of two Catholic priests (De Escalante 1995). Over the next eight decades, those who visited Heber Valley were primarily hunters and trappers employed by large fur trading companies in the eastern United States (Bancroft 1883; Embry 1996).

Heber Valley was relatively calm until the 1840s, when members of the Church of Jesus Christ of Latter-day Saints (hereafter referred to as the Church, per request of church leadership[1]) settled in the area. Due to political and religious differences, the group was persecuted and forced to leave their homes behind in the East and flee westward (Allen and Leonard 1992). Brigham Young, a prophet in the Church, led the first group of settlers into

the Salt Lake Valley in 1847, where they established a permanent settlement (Hunter 1949; Van Cott 1990; Allen and Leonard 1992). Shortly after, Brigham Young asked members of the Church to colonize the surrounding areas to find new farmland for incoming settlers.

In 1857, a group of the Church's men who were working at a sawmill in a canyon nearby decided to cross the top of the Wasatch Mountain Range into Heber Valley. It was believed that a paradise "lay nestled in the tops of the Wasatch Range", and when the group returned, it spread that there was a place with abundant farmland and water (Mortimer 1963). Settlers of Salt Lake were interested in developing a permanent settlement in Heber Valley, and in 1859, non-Native settlers came from Provo City, (about thirty miles southwest of Heber), to establish small ranches in the area (Mortimer 1963) to permanently settle. The initial group from Provo consisted of eleven men, three wagons, and several teams of oxen (Mortimer 1963). Upon arrival, they planted crops and built permanent homes (Mortimer 1963). With many reports commending the valley, the spring of 1860 saw a dramatic increase in families coming to Heber Valley. Since nearly all of the Church's settlers that established permanent residences in Heber were converted by, or personally knew apostle Heber C. Kimball, the residents chose in 1860 to name the settlement after him (Mortimer 1963; Van Cott 1990). Other communities, including Midway, Charleston, Daniel, and Center emerged over time.

### 3.2. The 2002 Salt Lake City Winter Olympics

In 1995, Salt Lake City was chosen as the host city for the 2002 Winter Olympics, and shortly after, in 1997, Soldier Hollow in Heber Valley was chosen to host Nordic skiing events (Cates 1997). Despite news of a bribery scandal surrounding the acquisition of the 2002 Salt Lake City Games, in 1999, major construction to prepare for the games began in the area, costing about USD 22 million (SLOC 2002).

With a population of around 16,000, 7338 housing units, and a median household income of USD 37,850 in 2002, a mega-event such as the Olympics was found to have significant impacts on the area's resources (Cope et al. 2015; Cope et al. 2021b; U.S. Census Bureau 2021). To accommodate the influx of people coming to Heber Valley, the Soldier Hollow Alternate Housing (SHAH) program constructed forty-two four-bedroom homes to house Olympic athletes and officials (Cope et al. 2015). Additionally, Heber officials created a "Western Experience" as entertainment for visitors between Olympic competitions, which included music, settler reenactments, and Native American displays (SLOC 2002). In order to reduce vehicle traffic in canyons leading into Wasatch County, the Salt Lake Organizing Committee (SLOC) came to an agreement with the Heber Valley Railroad to transport spectators to and from the Soldier Hollow venue (SLOC 2002). Analysis of survey data collected before, during, and after the 2002 Winter Olympics indicates mixed feelings amongst respondents regarding the impact of the games, and the data also suggest that increases in respondent satisfaction with and perceived desirability of the games were not sustained over time (Cope et al. 2015).

### 4. Materials and Methods

Our objectives are met via the joint analysis of survey data from the Heber Valley Community Survey (HVCS) and data obtained from a content analysis of *The Wasatch Wave*, a local newspaper—published weekly—serving the principal communities of the Heber Valley. First, we describe the survey and textual data, then discuss findings from the survey, our analysis of the newspaper data, and our findings from combining both data sources.

### 4.1. Survey Data

The HVCS—a repeated cross-sectional survey—consists of a baseline and six subsequent waves of data on resident sentiment and potential social impacts associated with community development and population growth. Administered by the Brigham Young University Survey Research Center (BYU SRC), the HVCS is a telephone-administered survey of residents of the five principal communities of Utah's Heber Valley (Heber, Midway,

Daniel, Charleston, and Center). Baseline data in the HVCS were collected in February 1999; four additional waves of data were collected annually—every February—between 2000 and 2003, with three additional waves of data gathered in February 2007, February 2012, and February 2018. Surveys were administered to individuals randomly selected each year from phone number databases through random dialing. The self-designated heads of households were the principal sampling unit. Sample sizes for the six waves of the HVCS ranged from 293 to 710, resulting in a combined final sample size of $N = 3285$ for the present study. As a repeated cross-sectional survey, the HVCS provides a unique opportunity to effectively measure change over time.

### 4.2. Textual Data

Textual data from *The Wasatch Wave* were drawn from its weekly publications from both of the two months (December and January) that preceded data collection associated with the HVCS. Newspapers were analyzed in the same years HVCS data were collected (1999–2003, 2007, 2012, and 2018). We elected to exclude textual data from the month of February because HVCS data were collected throughout the month of February, thereby allowing for the possibility that respondents at the beginning of the month would have been exposed to different community narrative topics than were respondents at the end of the month. From the selected weekly publications, we restricted our sample to editorials and letters to the editor, i.e., content from the opinion section. We omitted other articles and advertisements because ancillary analysis revealed them as beyond our current objectives of identifying how residents are experiencing community and social change.

### 4.3. Survey Data Modeling Approach

Our approach is based on Kasarda and Janowitz's (1974) systemic model of community, which has been extensively used to study social change (e.g., Brown et al. 2005; Brown et al. 1989; Krannich and Greider 1984). We aim to gauge the effectiveness of social change on community attachment and satisfaction, which we do by estimating their mean levels for each wave of the survey. At a conceptual level, significant changes in these means may be symptomatic of social changes triggered by the mega-event or community development projects, but they may also be indicative of compositional shifts in the population not caused by the mega-event. Accordingly, we replicated Kasarda and Janowitz's systemic model by controlling for length of residence, lifecycle stage, and social position, as well as race and sex.

#### 4.3.1. Survey Data Measures

These survey data measures and approaches build on the work of Cope et al. 2015. The dependent variables in our analysis are drawn from indicators established in the community disruption and general community well-being literature (e.g., Brown 1993; Brown et al. 2000; Park et al. 2015; McKnight et al. 2017). Specifically, we measure the effect of social change on community attachment and satisfaction. Our measure of *community attachment* is a scale based on two indicators gauging (1) the extent to which the respondent feels (s)he "fits" into the community and (2) how much the respondent believes (s)he has "in common" with other members of the community. These indicators are measured on a 5-point ordinal scale where higher scores are indicative of higher levels of attachment to the community. The alpha reliability of the attachment scale is good at 0.81. The scale is calculated as the mean of the scores on the two variables. *Community satisfaction* is also measured with a scale based on two indicators (see Brown 1993; Brown et al. 2000; Park et al. 2015; McKnight et al. 2017). The first considers the extent to which respondents' present community matched their ideal community, while the second assesses how satisfied residents are with living in their community. These are both measured on 5-point ordinal scales where higher scores indicate higher levels of satisfaction. The alpha reliability for the satisfaction scale is good at 0.85. The scale is calculated as the mean of the scores on the two variables. An additional measure of community satisfaction, which we label

*community desirability*, is also included in our analysis. In terms of community desirability, respondents were asked to indicate if their community has become "less desirable, stayed about the same, or become more desirable as a place to live" in the five years preceding the survey. "Less desirable" was coded as 1, "stayed about the same" was coded as 2, and "more desirable" as 3.

Our key independent variable in our analysis is social change over time. We do this by conceptualizing the multiple survey years as a measure of community change as it relates to our dependent variables. As such, we pooled seven waves of cross-sectional survey data and created dummy variables for each wave (holding the 1999 baseline wave as the reference). When included in the models, the coefficient for each dummy variable represents a measure of the difference in the dependent variable between the first year of the survey and each of the subsequent years of the survey.

Our models also include a range of control variables, as justified by Kasarda and Janowitz's (1974) elaboration of the systemic model of community. Specifically, Kasarda and Janowitz argue that the primary determinants of attachment to a local community are length of residence, lifecycle stage, social position, and social ties. *Length of residence* is measured as the proportion of a respondent's life spent residing in the community (i.e., the quotient of the number of years a resident has lived in the community divided by that resident's age). Most studies use the raw number of years a resident has resided in the community to measure the length of residence but doing so conflates the effects of length of residence and age, which we also control for in our analysis (see Flaherty and Brown 2010; Magno and Dossena 2020). In our data, with rounding, length of residence ranges from 0 to 1. Three variables are used to control for the effect of lifecycle stage: *age* (measured as a continuous-level variable), *number of children living in the respondent's household* (an ordinal measure ranging from "0 children" to "6 or more children"), and *marital status* (coded as 1 for respondents who are married or widowed and 0 for all others). To gauge a respondent's *social position,* we control for the number of years of schooling they reported having completed. Additionally, we include dummy variables for *race* (1 = white) and *sex* (1 = female). To control for *social ties*, respondents were asked "about what percentage of adults in this community would you say that you know on a first name basis?" Responses were coded on a 1–4 scale, where 1 indicated 0 to 24% and 4 indicated 75 to 100%. Descriptive statistics for all variables used in the analysis are shown in Table 1.

**Table 1.** Descriptive Statistics.

| | 1999 (N = 337) | | 2000 (N = 364) | | 2001 (N = 367) | | 2002 (N = 337) | | 2003 (N = 293) | |
|---|---|---|---|---|---|---|---|---|---|---|
| | Mean (Percent) | SD | Mean (Percent) | SD | Mean (Percent) | SD | Mean (Percent) | SD | Mean (Percent) | SD |
| **Dependent Variables** | | | | | | | | | | |
| Attachment | 3.711 | 0.92 | 3.753 | 0.923 | 3.726 | 0.918 | 3.804 | 0.914 | 3.858 | 0.913 |
| Satisfaction | 3.912 | 0.91 | 3.956 | 0.840 | 3.876 | 0.925 | 4.074 | 0.834 | 4.065 | 0.836 |
| Desirability | 1.855 | 0.83 | 1.780 | 0.827 | 1.809 | 0.831 | 2.187 | 0.844 | 2.027 | 0.806 |
| **Independent Variables** | | | | | | | | | | |
| Length of Res | 0.437 | 0.34 | 0.457 | 0.342 | 0.427 | 0.359 | 0.361 | 0.316 | 0.365 | 0.330 |
| Age | 44.855 | 14.48 | 47.885 | 16.456 | 48.144 | 17.192 | 47.332 | 15.786 | 51.925 | 18.518 |
| Children | 1.804 | 1.73 | 1.684 | 1.664 | 1.619 | 1.565 | 1.641 | 1.690 | 1.611 | 1.716 |
| Married or widowed | 0.807 | | 0.816 | 0.388 | 0.850 | 0.357 | 0.855 | 0.353 | 0.843 | 0.364 |
| Education | 3.923 | 1.678 | 4.047 | 1.803 | 3.978 | 1.713 | 4.389 | 1.722 | 4.399 | 1.781 |
| White | 0.970 | | 0.962 | | 0.937 | | 0.961 | | 0.952 | |
| Female | 0.427 | | 0.448 | | 0.414 | | 0.427 | | 0.406 | |
| Social Ties | 1.789 | 0.979 | 1.783 | 0.936 | 1.790 | 0.916 | 1.807 | 0.977 | 1.703 | 0.927 |
| **Textual Data Variables** | | | | | | | | | | |
| Development | −0.737 | | −0.246 | | −0.913 | | −0.268 | | 1.001 | |
| Growth | −0.814 | | 0.491 | | 0.246 | | −3.925 | | 0.423 | |
| Sentiment | −1.224 | | −0.406 | | −0.257 | | −0.297 | | 0.895 | |

**Table 1.** *Cont.*

| | 2007 (N = 565) | | 2012 (N = 312) | | 2018 (N = 710) | | Full Sample (N = 3285) | |
|---|---|---|---|---|---|---|---|---|
| | Mean (Percent) | SD | Mean (Percent) | SD | Mean (Percent) | SD | Mean (Percent) | SD |
| Dependent Variables | | | | | | | | |
| Attachment | 3.840 | 0.883 | 4.043 | 0.832 | 4.923 | 1.219 | 4.056 | 1.082 |
| Satisfaction | 3.999 | 0.866 | 4.362 | 0.753 | 5.454 | 1.301 | 4.334 | 1.140 |
| Desirability | 1.912 | 0.875 | 2.202 | 0.766 | 1.921 | 0.817 | 1.948 | 0.839 |
| Independent Variables | | | | | | | | |
| Length of Res | 0.412 | 0.348 | 0.468 | 0.315 | 0.358 | 0.325 | 0.405 | 0.337 |
| Age | 53.929 | 17.376 | 58.426 | 14.848 | 57.070 | 15.478 | 51.933 | 16.906 |
| Children | 1.432 | 1.658 | 0.974 | 1.509 | 2.106 | 0.819 | 1.658 | 1.540 |
| Married or widowed | 0.908 | 0.289 | 0.926 | 0.262 | 0.870 | 0.336 | 0.863 | 0.344 |
| Education | 4.577 | 1.755 | 4.638 | 1.793 | 5.086 | 1.794 | 4.465 | 1.804 |
| White | 0.965 | | 0.971 | | 0.972 | | 0.963 | |
| Female | 0.375 | | 0.304 | | 0.582 | | 0.439 | |
| Social Ties | 1.596 | 0.853 | 1.801 | 0.917 | 1.437 | 0.738 | 1.675 | 0.897 |
| Textual Data Variables | | | | | | | | |
| Development | 3.125 | | −1.777 | | −1.119 | | −0.016 | |
| Growth | −0.802 | | 0.751 | | 1.836 | | 0.028 | |
| Sentiment | 0.467 | | 3.105 | | −1.114 | | −0.015 | |

Sources: HVCS and *The Wasatch Wave.*

### 4.3.2. Survey Data Modeling Strategy

We specify ordinary least squares (OLS) regression models to predict continuously measured scales of community attachment and community satisfaction. We take advantage of the ordinal nature of our measure of community desirability and, conversely, specify a model using ordinal logistic regression. For models predicting community desirability, we present odds ratios[2].

### 4.4. Analytic Approach to Textual Data

To gain insight into the narratives used by the residents of Heber Valley to describe social change in their community, textual data were analyzed in two stages. First, we inductively coded the selected texts for emergent patterns and themes (Glaser and Strauss 1967). Throughout this stage of open coding, from the 660 textual samples (311 main page; 349 opinion page) we identified twenty-one broad and overlapping codes: during this stage of coding, each researcher wrote memos making note of patterns within the coding categories. During the second stage, the coders returned to the texts again to further develop sub-codes within each of the preliminary categories; this approach is referred to by Charmaz (2006) as "fracturing" the data. At this point, we also coded for the tone of each text on a 1–5 scale (1 = *negative*, 2 = *somewhat negative*, 3 = *neutral*, 4 = *somewhat positive*, 5 = *positive*). We then organized these data using seven categories that formed the foundation of our emergent analysis:

1.  narratives about community sentiment,
2.  discussion of the 2002 Winter Olympic games,
3.  concerns of development and planning,
4.  expressed concerns about "growing pains" associated with changing community characteristics,
5.  economic and business concerns,
6.  conversations regarding elected officials, and
7.  reputation of the local public schools.

This stage of the content analysis represented a transition from open coding to one where we developed robust integrative categories for understanding issues discussed by the residents of Heber Valley. Each of the categories of the integrative narrative of change is discussed in greater detail below.

## 5. Results

### 5.1. Textual Analysis of The Wasatch Wave

Here, we briefly discuss the results of our content analysis of *The Wasatch Wave*, especially looking for evidence of narratives of social change. Specifically, we describe seven categories that emerged from our analysis: (1) narratives about community sentiment, (2) discussion of the 2002 Winter Olympic games, (3) concerns of development and planning, (4) expressed concerns about "growing pains" associated with changing community characteristics, (5) economic and business concerns, (6) conversations regarding elected officials, and (7) reputation of the local public schools.

*Community Sentiment* is a category that tries to capture how the people of Heber are feeling about their community. This included any articles that dealt with the emotions of these community members that were anywhere on the scale of positive to negative. For example, some of the articles were about volunteering, youth athletics, historic preservation, and other local activities. For example, the sentiment, "This Christmas ballet has become an exceptional tradition in our small town" (16 December 1998) was coded as positive (5), "Winters in Wasatch County can be dangerous, yet some of the most beautiful in the West. Stay safe and enjoy Utah's splendor" (14 December 2011) was coded as neutral (3), and "Wouldn't it be better to be known as the best fans in the state?" (13 January 1999) was coded as negative (1).

The *Olympics* were held in 2002, but there was a lot of talk about the games for a couple of years before they occurred as well as for a couple of years after they occurred. This category included any stories that focused on the Olympic Games. This included talk about the use of the facilities, tickets, volunteers, and the Special Olympics, whether positive or negative. Examples include, "The torrid Salt Lake Olympic Scandal, that caused uproar across the world, is now being blamed on the Mormon Church" (27 January 1999), which was coded as negative (1), "Mayor Adams, please remove the piles of snow from Main and give us more room for parking. We need the people to stop and shop" (19 December 2001), which was coded as neutral (3), and "We know we are not Salt Lake; we know we are not Park City, and Provo and Ogden are always letting us know we are not them ... what we intend to do is be ourselves" (30 January 2002), which was coded as positive (5).

*Overall Development*: Over the time frame of our sample, Heber grew from just a small rural town to a known place on the map because of the Winter Games. During this time, lots of new houses and buildings were going up to try to support that growth. This category was filled with articles that dealt with zoning laws, water, crime, roads, and other issues that relate to this growth. Some examples are, "I love it here but perhaps there are some areas where change might be healthy" (31 January 2007), which was coded as positive (5), "This is the county's first performance audit. The anticipated audit should be viewed as a positive step that may reveal possible financial mismanagement and whether state laws or local policies have been broken" (22 December 1999), which was coded as neutral (3), and "When you have to wait three hours to use the facilities and then be faced with [the] rudeness of our law enforcement, it makes you wonder what our tax money is really going for" (27 December 2000), which was coded as negative (1).

*Growing Pains*: When it comes to growth, not everything that comes with it will make everyone in the community happy. This category includes articles that focus on the negative consequences of growth. Articles talked about tax money and where it was going, pollution and wildlife loss, gas and electric companies, and where new roads should be built. For example, residents said, "I urge the Wasatch-Cache land managers to explore alternatives to wilderness that still conserve the primitive qualities of the Weber Lakes/Mt. Watson while continuing to allow a prescription of over-the-snow vehicles during the winter" (8 December 1999), "Heber already has an air pollution problem, and an asphalt plant can only make it worse. That's why the asphalt plant must be stopped-it can only make a bad situation worse" (20 December 2000), and "If Heber Light is so affluent, how about rolling back that $12 additional monthly fee that was socked to us about 18 months or so ago" (11 January 2012), which was coded as negative (1).

*Local Economy*: The local economy is a topic that is of huge importance to the people who reside in Heber. There were many articles that dealt with concerns about or praise of businesses or people that affected Heber's economy. Throughout the time frame of this research, the local economy played a critical role in the community story. Some comments that were coded as negative include (1) "As you read this you have about a month-one month left to let the Winterton's know how much they have added to a quiet little mountain town and valley" (20 December 2000) and "We cannot 'plan' or 'legislate' Midway into commercial prosperity" (29 January 2003). Further, comments that were coded as positive included (5) "'Cha-ching.' The sound of cash registers ringing up sales translates into a healthy community" (4 December 2002) and "Why all the fear? Fear freezes the mind, imagination, and creativity. If Heber City had these big box stores, whoever they are, there would be folks driving from all over the area to shop in Heber City" (24 January 2007).

*Elected Officials*: Every town, city, or county has some form of government that helps dictate and uphold the laws of that area. In the case of Heber, the elected officials continually dealt with the laws that affected the growth that was occurring. This category included any articles that mentioned the elections, their community involvement, and any distrust the people felt about the officials in office. Many articles were coded as negative (1). Some comments from these articles include, "It's a shame [that] citizens don't give a damn where or how their tax dollars are spent" (16 December 1998), "The days of secret meetings, unqualified closed-door sessions and morning coffee at the local breakfast shop where some subjects just 'happen' to come up is at an end" (4 January 2012), and "Now comes, the choice and accountability payment for your choices. You will not be thought of as honest people anymore, reputations will be tarnished. It is so sad" (18 January 2012).

*Reputation of Local Public Schools*: With growth comes more people, which means more children. This means that public schools had to follow the same growth pattern. The effects schools have on a community's legacy have been documented (see Versteeg 1993; Bauch 2001; Lasater et al. 2022). This category included any articles that mention Heber's schools, whether positively or negatively. Many articles talked negatively about the budgets and teachers, but they also mentioned the students' academic and athletic achievements as positive features of the community. For example, the comment, "Just a note to say thanks to you for allowing us the opportunity to work with your children at Midway Elementary School" (15 December 1999) was coded as positive (5), "By instituting a closed campus, pedestrian congestion across Main Street would be eliminated" (1 December 1999) was coded as neutral (3), and "In conclusion, I invite the Heber Valley, The Wasatch Wave newspaper, and the surrounding areas of Wasatch High School to support Wasatch High School's rule of modesty" (11 January 2021) was coded as negative (1).

*5.2. Survey Data*

To verify our findings of narratives of social change in our analysis of *The Wasatch Wave*, we also used survey data from the same time period to analyze community change. Analysis of the survey data also allowed us to have a more nuanced discussion about social change over time by separating the results from survey years. Table 2 presents the regressions of community attachment on social change and the control variables. The first model, equivalent to an analysis of variance, is effectively a bivariate model that uses dummy variables for survey year to measure changes in the mean level of community attachment across time. The results show that the mean level of community attachment was significantly higher in 2012 and 2018 than in 1999.

In the second model, we included the full range of control variables. In the presence of the control variables, the previously modeled effect showing that the mean level of community attachment was significantly higher in 2012 and 2018 than in 1999 persists. With regard to the other variables included in model 2, we find that older respondents, those reporting having more children living in the home, being married or widowed, having a higher level of education, and having more social ties are significantly associated with greater levels of community attachment. Interestingly, we did not find a significant

association between length of residence and community attachment. There is extensive literature on the effects of length of residence on social ties and attachment, and the existence of regular patterns that hold between those variables are well established (e.g., Kasarda and Janowitz 1974; Flaherty and Brown 2010). Longer-term residence is generally associated with more local social ties and higher levels of attachment. In our data, longer-term residents' levels of attachment are not significantly different from those of shorter-term residents'. However, when the variable for "social ties" is removed from the model, the effect of length of residence on attachment becomes positive and significant.

**Table 2.** Change in Community Attachment Over Time.

| | Model 1 | | | Model 2 | | |
|---|---|---|---|---|---|---|
| | **b** | | **SE** | **b** | | **SE** |
| 1999 (reference) | | | | | | |
| 2000 | 0.040 | | 0.074 | 0.031 | | 0.071 |
| 2001 | 0.019 | | 0.074 | 0.002 | | 0.071 |
| 2002 | 0.090 | | 0.075 | 0.068 | | 0.072 |
| 2003 | 0.137 | | 0.078 | 0.124 | | 0.075 |
| 2007 | 0.128 | | 0.067 | 0.109 | | 0.065 |
| 2012 | 0.310 | *** | 0.077 | 0.226 | ** | 0.075 |
| 2018 | 1.210 | *** | 0.065 | 1.189 | *** | 0.065 |
| Length of Residence | | | | 0.081 | | 0.052 |
| Age | | | | 0.005 | *** | 0.001 |
| Children | | | | 0.061 | *** | 0.012 |
| Marital Status | | | | 0.344 | *** | 0.049 |
| Education | | | | 0.019 | * | 0.010 |
| White | | | | 0.104 | | 0.086 |
| Male | | | | −0.041 | | 0.034 |
| Social Ties | | | | 0.259 | *** | 0.019 |
| Constant | 3.712 | *** | 0.053 | 2.424 | *** | 0.126 |
| N | | 3285 | | | 3285 | |
| $R^2$ | | 0.182 | | | 0.258 | |
| Adj. $R^2$ | | 0.181 | | | 0.254 | |
| F | | 105.06 | | | 76.23 | |
| P | | <0.001 | | | <0.001 | |

Notes: * $p < 0.05$, ** $p < 0.01$, *** $p < 0.001$. Source: HVCS.

Table 3 presents the regressions of community satisfaction on social change and the control variables. In the unconditional model, we show that the mean level of community satisfaction is significantly lower in 1999 than in 2002, 2003, 2012, and 2018. The effects for 2002 and 2003 disappear, however, in the conditional model where the only difference in community satisfaction over time appears in 2012 and 2018 compared to the 1999 baseline. With regard to the control variables, as expected, we find that older respondents, married or widowed respondents, those with higher levels of education, male respondents, and those reporting greater numbers of social ties are significantly likely to be more satisfied with their community compared to others. Interestingly, our models show that women on average reported marginally significant lower levels of community satisfaction than men. Surprisingly, the results also show that, in our models, longer-term residents report lower levels of community satisfaction than do short-term residents. To our knowledge, not enough research has been conducted to explore the relationship between satisfaction and length of time of residence. Consequently, we lack theoretical guidance to interpret this finding. However, to further explore this finding, we removed social ties from the model and the pattern remained the same: longer-term residents report lower levels of community satisfaction.

**Table 3.** Change in Community Satisfaction Over Time.

| | Model 1 | | | Model 2 | | |
|---|---|---|---|---|---|---|
| | **b** | | **SE** | **b** | | **SE** |
| 1999 (reference) | | | | | | |
| 2000 | 0.044 | | 0.073 | 0.034 | | 0.072 |
| 2001 | −0.036 | | 0.073 | −0.063 | | 0.072 |
| 2002 | 0.162 | * | 0.075 | 0.108 | | 0.073 |
| 2003 | 0.152 | * | 0.077 | 0.095 | | 0.076 |
| 2007 | 0.087 | | 0.067 | 0.028 | | 0.066 |
| 2012 | 0.450 | *** | 0.076 | 0.348 | ** | 0.077 |
| 2018 | 1.541 | *** | 0.064 | 1.476 | *** | 0.066 |
| Length of Residence | | | | −0.256 | | 0.053 |
| Age | | | | 0.005 | *** | 0.001 |
| Children | | | | 0.022 | *** | 0.012 |
| Marital Status | | | | 0.268 | *** | 0.050 |
| Education | | | | 0.023 | * | 0.010 |
| White | | | | 0.062 | | 0.088 |
| Male | | | | −0.076 | | 0.034 |
| Social Ties | | | | 0.158 | *** | 0.020 |
| Constant | 3.912 | *** | 0.053 | 3.141 | *** | 0.128 |
| N | | 3285 | | | 3285 | |
| $R^2$ | | 0.278 | | | 0.311 | |
| Adj. $R^2$ | | 0.276 | | | 0.308 | |
| F | | 180.18 | | | 98.47 | |
| P | | <0.001 | | | <0.001 | |

Notes: * $p < 0.05$, ** $p < 0.01$, *** $p < 0.001$. Source: HVCS.

Table 4 displays the results of the ordinal logistic regression models for community desirability. In the unconditional model, the odds of finding the community more desirable are estimated to be significantly higher in 2002, 2003, and 2012 than in 1999. For example, the odds of reporting that the community is more desirable were nearly 2.1 times as high in 2002 as in 1999, were 55% higher in 2003 than in 1999, and approximately 2.2 times as high in 2012 as in 1999. This pattern persists when other controls are introduced. With regard to the control variables, we find that among longer-term residents, older residents, white residents, and male residents, the odds of finding the community more appealing are estimated to be significantly lower when compared to others.

**Table 4.** Change in Community Desirability Over Time.

| | Model 1 | | | Model 2 | | |
|---|---|---|---|---|---|---|
| | **OR** | | **SE** | **OR** | | **SE** |
| 1999 (reference) | | | | | | |
| 2000 | 0.851 | | 0.119 | 0.896 | | 0.128 |
| 2001 | 0.921 | | 0.128 | 0.907 | | 0.129 |
| 2002 | 2.199 | *** | 0.315 | 2.040 | *** | 0.298 |
| 2003 | 1.492 | ** | 0.216 | 1.437 | * | 0.214 |
| 2007 | 1.132 | | 0.145 | 1.197 | | 0.158 |
| 2012 | 2.117 | *** | 0.302 | 2.506 | *** | 0.374 |
| 2018 | 1.190 | | 0.144 | 1.308 | * | 0.170 |
| Length of Residence | | | | 0.262 | *** | 0.028 |
| Age | | | | 0.986 | *** | 0.002 |
| Children | | | | 0.956 | | 0.023 |
| Marital Status | | | | 0.997 | | 0.099 |
| Education | | | | 1.053 | ** | 0.020 |
| White | | | | 0.529 | *** | 0.095 |
| Male | | | | 0.819 | ** | 0.055 |
| Social Ties | | | | 1.169 | *** | 0.045 |
| N | | 3285 | | | 3285 | |
| $R^2$ | | 0.013 | | | 0.050 | |
| Adj. $R^2$ | | 93.700 | | | 361.680 | |
| F | | <0.001 | | | <0.001 | |
| P | | 3285 | | | 3285 | |

Notes: * $p < 0.05$, ** $p < 0.01$, *** $p < 0.001$. Sources: HVCS and *The Wasatch Wave*.

### 5.3. Combining Textual and Survey Data

Here, we combine the analysis presented above to assess the validity of the two approaches against each other. To do so, we coded the tone of each text sampled from *The Wasatch Wave*. Specifically, we coded tone in terms of the following 1–5 scale: 1 = *negative*, 2 = *somewhat negative*, 3 = *neutral*, 4 = *somewhat positive*, and 5 = *positive*. We then combined the textual data with the survey data and used principal component factor analysis with varimax rotation to reduce the seven categories identified in the content analysis to three factors representing independent dimensions of the narratives of social change. Principal components analysis was chosen in order to eliminate any multicollinearity that might exist in the measures. The principal component analysis, see Table 5, resulted in a three-factor solution that accounted for nearly 88.93% of the total variance found in the seven dimensions identified in the narratives of social change. The factors reflected perceptions of the positive and negative aspects of development, growth, and sentiment. Factor names are based on the characteristics of included variables, specifically a dominant variable. Each of these components is discussed in greater detail below.

**Table 5.** Principal Component Analysis of Textual Data.

| Factor Name | Percent Variance Explained | Text Variable | Factor Loading |
|---|---|---|---|
| Development | 36.25% | Overall Development | 0.46 |
| | | Local Economy | 0.59 |
| | | Elected Officials | −0.64 |
| Growth | 30.43% | Olympics | −0.61 |
| | | Growing Pains | 0.51 |
| | | Local Public Schools | 0.55 |
| Sentiment | 21.25% | Community Sentiment | 0.80 |
| Total variance among items: | | | 87.93% |

*Development*: The first factor is noticeably related to discussions surrounding community development. Specifically, overall development and local economy load positively on this factor, while elected officials loads negatively. This factor accounts for 36.25% of the variance shown among variables in the model.

*Growth*: The second factor is indicative of growth in general as well as growth related to the Olympics. Olympics loads negatively while growing pains and local public schools load positively. This factor accounts for 30.43% of the variance shown among variables in the model.

*Sentiment*: The third and final factor corresponds to concerns over community sentimentality and accounts for 21.25% of the variance shown among variables in the model. Community sentiment loads positively.

The aforementioned factors are indicative of community change in that they exemplify the relations between community actors occupying a particular place in time and the inherited stories related to that location and time (e.g., Phillips 2002; Calhoun 1991). As such, we take advantage of the nature of these measures to assess the relationship between the narratives of social change and the previously discussed measures of community attachment, community satisfaction, and community desirability. To that end, we specify regression models similar to those previously discussed, substituting the principal component measures of community change for the dummy variables used to measure the difference in the dependent variable between the first year of the survey and each of the subsequent years of the survey.

Table 6 presents the regressions of community attachment, community satisfaction, and community desirability on social change using the factors of social change and the control variables. Following the same modeling strategies previously used, the first model is effectively a bivariate model that uses the factors of social change to measure variations in the mean level of community attachment. Across the different outcomes presented

in Table 6, the results show a significant positive relationship between growth and the outcome variables. This same pattern persists when controlling for other variables. Further, development and sentiment have a significant negative relationship with community attachment and community satisfaction variables, even when including control variables. However, this pattern changes when predicting desirability. All three factors have a significant positive relationship with community desirability, including when control variables are added to the model.

**Table 6.** Change in Community Sentiments Predated with Textual Factors.

| | Community Attachment | | | | | | | | Community Satisfaction | | | | | | | |
| --- | --- | --- | --- | --- | --- | --- | --- | --- | --- | --- | --- | --- | --- | --- | --- | --- |
| | Model 1 | | | Model 2 | | | | | Model 3 | | | Model 4 | | | |
| | b | | SE | b | | SE | | | b | | SE | b | | SE | |
| Development | −0.031 | * | 0.012 | −0.030 | * | 0.012 | | | −0.061 | *** | 0.013 | −0.067 | *** | 0.012 | |
| Growth | 0.198 | *** | 0.013 | 0.180 | *** | 0.013 | | | 0.241 | *** | 0.014 | 0.215 | *** | 0.014 | |
| Sentiment | −0.101 | *** | 0.015 | −0.115 | *** | 0.015 | | | −0.126 | *** | 0.015 | −0.139 | *** | 0.015 | |
| Length of Residence | | | | 0.052 | | 0.055 | | | | | | −0.279 | *** | 0.057 | |
| Age | | | | 0.010 | *** | 0.001 | | | | | | 0.010 | *** | 0.001 | |
| Children | | | | 0.083 | *** | 0.012 | | | | | | 0.051 | *** | 0.013 | |
| Marital Status | | | | 0.338 | *** | 0.052 | | | | | | 0.263 | *** | 0.054 | |
| Education | | | | 0.049 | *** | 0.010 | | | | | | 0.060 | *** | 0.010 | |
| White | | | | 0.137 | | 0.091 | | | | | | 0.103 | | 0.095 | |
| Male | | | | −0.031 | | 0.035 | | | | | | −0.052 | | 0.037 | |
| Social Ties | | | | 0.232 | *** | 0.020 | | | | | | 0.127 | *** | 0.021 | |
| Constant | 4.048 | *** | 0.018 | 2.380 | *** | 0.124 | | | 4.324 | *** | 0.018 | 3.042 | *** | 0.130 | |
| N | 3285 | | | 3285 | | | | | 3285 | | | 3285 | | | |
| $R^2$ | 0.099 | | | 0.050 | | | | | 0.145 | | | 0.202 | | | |
| Adj. $R^2$ | 0.098 | | | 361.680 | | | | | 0.144 | | | 0.199 | | | |
| F | 119.45 | | | 67.43 | | | | | 185.64 | | | 75.23 | | | |
| P | <0.001 | | | <0.001 | | | | | <0.001 | | | <0.001 | | | |
| | Community Desirability | | | | | | | | | | | | | | | |
| | Model 5 | | | Model 6 | | | | | | | | | | | | |
| | OR | | SE | OR | | SE | | | | | | | | | | |
| Development | 0.930 | ** | 0.021 | 0.926 | ** | 0.021 | | | | | | | | | | |
| Growth | 0.908 | *** | 0.022 | 0.927 | ** | 0.023 | | | | | | | | | | |
| Sentiment | 1.160 | *** | 0.030 | 1.187 | *** | 0.033 | | | | | | | | | | |
| Length of Residence | | | | 0.256 | *** | 0.028 | | | | | | | | | | |
| Age | | | | 0.989 | *** | 0.002 | | | | | | | | | | |
| Children | | | | 0.967 | | 0.023 | | | | | | | | | | |
| Marital Status | | | | 0.990 | | 0.098 | | | | | | | | | | |
| Education | | | | 1.076 | *** | 0.021 | | | | | | | | | | |
| White | | | | 0.543 | ** | 0.098 | | | | | | | | | | |
| Male | | | | 0.826 | ** | 0.056 | | | | | | | | | | |
| Social Ties | | | | 1.154 | *** | 0.045 | | | | | | | | | | |
| N | 3825 | | | 3285 | | | | | | | | | | | | |
| Pseudo-$R^2$ | 0.007 | | | 0.044 | | | | | | | | | | | | |
| LR Chi$^2$ | 50.810 | | | 314.730 | | | | | | | | | | | | |
| P | <0.001 | | | <0.001 | | | | | | | | | | | | |

Notes: * $p < 0.05$, ** $p < 0.01$, *** $p < 0.001$. Source: HVCS and The Wasatch Wave.

## 6. Discussion

The goal of the present study was to analyze narratives of social change. Specifically, we (1) explore how Heber Valley residents' narratives change as a result of preparing for, participating in, and recovering from the Olympics, (2) verify these findings using survey data gathered during the same time period, and (3) examine how changes in residents' narratives in Heber Valley impacted the subjective evaluation of community. Our dependent variables were community attachment, community satisfaction, and community

desirability. As established in the literature on community, these variables allow us to measure social change (see Brown 1993; Brown et al. 2000; Park et al. 2015; McKnight et al. 2017).

We used a mixed methods approach to measure changes in community sentiment over time. First, we analyzed articles from the months of December and January in a Heber Valley newspaper, *The Wasatch Wave*, published from 1999 through 2003, as well as from 2007, 2012, and 2018. Seven categories emerged from our analysis: (1) narratives about community sentiment, (2) discussion of the 2002 Winter Olympic games, (3) concerns of development and planning, (4) expressed concerns about "growing pains" associated with changing community characteristics, (5) economic and business concerns, (6) conversations regarding elected officials, and (7) reputation of the local public schools. Through our analysis, we found that residents' narratives did indeed change as a result of preparing for, participating in, and recovering from the Olympics as seen through the seven categories that emerged.

In addition, we also used survey data collected in Heber Valley from 1999 to 2003, with additional waves collected in 2007, 2012, and 2018. The survey data were used to verify our findings from our analysis of the newspaper, as well as provide a more nuanced picture of changes in community over time. In our analysis of the survey data, we found that when including control variables, the mean level of community attachment is significantly higher in 2012 and 2018 than in 1999. This indicates that community growth has influenced community attachment over time. With regard to the other variables included in predicting community satisfaction, we find that older respondents, those reporting having more children living in the home, being married or widowed, having a higher level of education, and having more social ties to be significantly associated with greater levels of community attachment. Interestingly, we did not find a significant association between length of residence and community attachment. However, when the variable of social ties is removed from the model, the effect of length of residence on attachment becomes positive and significant.

We also used survey data to predict community satisfaction. Similar to community attachment, when all control variables were included, satisfaction was significantly higher in 2012 and 2018 compared to 1999. Again, we find that community satisfaction has increased over time as the community has changed. With regard to the control variables, as expected, we find that older respondents, married or widowed respondents, those with higher levels of education, male respondents, and those reporting greater numbers of social ties are significantly likely to be more satisfied with their community compared to others. Interestingly, our models show that women on average reported marginally significant lower levels of community satisfaction than men. Even more surprising, though, the results also show that, in our models, longer-term residents report lower levels of community satisfaction than short-term residents. To our knowledge, not enough research has been conducted to explore the relationship between satisfaction and length of time of residence. We encourage more research to be conducted investigating this relationship.

Finally, we also predicted community desirability using survey data. In the conditional model, the odds of finding the community more desirable are estimated to be significantly higher in 2002, 2003, and 2012 than in 1999. With regard to the control variables, we find that odds of finding the community more desirable are estimated to be significantly lower among longer-term residents, older residents, white residents, and male residents when compared to others.

Last, the newspaper articles were coded on a 1–5 scale, where 1 indicates negative opinions or comments, 3 indicates neutral opinions or comments, and 5 indicates positive opinions or comments. Three different factors were then created from these data: (1) development, which includes the themes of overall development, local economy, and elected officials; (2) growth, which includes the Olympics, growing pains, and local public school categories; and (3) sentiments, which only includes the community sentiment category. Then, we used the factors to predict community attachment, satisfaction, and desirability. We found a significant positive relationship between growth and the outcome

variables, even when control variables are included in the model. Further, development and sentiment have a significant negative relationship with the community attachment and community satisfaction variables, even when including control variables. However, this pattern changes when predicting desirability. All three factors have a significant positive relationship with community desirability, including when control variables are added to the model. Overall, the factors created by the categories found in our analysis of the newspapers consistently predict feelings towards community at significant levels, leading us to conclude that narratives around community sentiment were associated with how residents subjectively evaluated their community through surveys.

While the present study contributes to the community literature, there are limitations to our research. First, newspapers, especially small-town newspapers, are disappearing and being replaced with other forms of communication, such as social media pages. In the future, it is important to use more current forms of communication within communities to analyze changes taking place. Second, we focused on place-based community instead of other possible conceptualizations. As outlined in Bradshaw (2008), place-based community requires a common geographical location but may lack social solidarity and involvement. This focus on place-based community may limit our understanding, and we encourage future researchers to investigate global networks that may foster or hinder community sentiments. Third, variables that may be relevant were excluded from the models. For example, due to coding issues with the 2012 survey data, religion was excluded from our analyses, which may be a significant predictor of community sentiment because of the region's large population of members of the Church of Jesus Christ of Latter-day Saints. Last, since survey data were collected across almost 30 years and conducted by different researchers, there are some variations in question construction that could introduce measurement error into our data.

## 7. Conclusions

The findings presented in this paper add to the growing body of research conducted to investigate changes in a community that are associated with community growth. The present study confirms well-documented findings that community does indeed change with growth. Our research contributes to the community literature by testing that existing data that can be accessed through unobtrusive measures, such as newspapers or community groups on social media, also show similar patterns to what has been found through survey data; growth over time is associated with changes in how residents feel about their community. We encourage future research on community change to continue to test this finding by comparing existing data to survey data in other areas.

**Author Contributions:** Conceptualization, M.R.C.; methodology, M.R.C., S.R.S., and C.W; software, M.R.C., and H.M.J.; validation, M.R.C.; formal analysis, M.R.C., and H.M.J.; data curation, M.R.C. and H.M.J.; writing—original draft preparation, M.R.C., H.M.J., G.L.A., H.Z.H., and E.L.-M.; writing— review and editing, M.R.C. and H.M.J.; visualization, M.R.C. and H.M.J.; supervision, M.R.C., S.R.S. and C.W.; project administration, M.R.C.; funding acquisition, M.R.C., S.R.S. and C.W. All authors have read and agreed to the published version of the manuscript.

**Funding:** Data collection efforts were funded in part by two sources internal to Brigham Young University: The Charles Redd Center for Western Studies, and the College of Family, Home, and Social Sciences.

**Institutional Review Board Statement:** Survey data used in this paper were gathered as part of a study that was reviewed and approved by the Brigham Young University Institutional Review Board.

**Informed Consent Statement:** Informed consent was obtained from all subjects involved in the study.

**Data Availability Statement:** The datasets generated during and/or analyzed during the current study are available from the corresponding author on reasonable request.

**Acknowledgments:** The authors thank the students in the BYU Communities Studies Lab for help during the data collection and curation phases of the project.

**Conflicts of Interest:** The authors declare no conflict of interest.

## Notes

1  See Style Guide of the Church: https://newsroom.churchofjesuschrist.org/style-guide (accessed on 21 November 2022). Cope et al. (2021a, p. 7) note the following: Researchers often use the word "Mormon" in reference to various religious and cultural groups historically linked to the Latter Day Saint movement founded in the late 1820s in upstate New York by Joseph Smith Jr. However, the term has most frequently been used as slang moniker for members of largest sect: The Church of Jesus Christ of Latter-day Saints, which is currently headquartered in Salt Lake City, UT, USA. While the original meaning of the term "Mormon" was descriptive for individuals who espoused belief in the Book of Mormon as a volume of scripture, the word has also historically also been used in a derogatory manner. Indeed, the Salt Lake-based denomination has experienced a complicated history with the term—oscillating between embracing and distancing—with the current attitude espoused by ecclesiastical leaders that the term is pejorative. For this reason, and a desire to use the preferred nomenclature of our target population, in this paper, we restrain from referring to members of the church, or the church itself, as "Mormon".

2  Odds ratios below 1.0 indicate a negative relationship between the dependent and independent variable, odds ratios of 1.0 indicate no relationship, and odds ratios above 1.0 indicate a positive relationship.

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
