# Peer review of "It Is Home: Perceptions, Community, and Narratives about Change"

_socsci, doi:10.3390/socsci12020081_

Round 1

Reviewer 1 Report

The manuscript focuses on narratives about their community and community stories in Utah’s Heber Valley and social change. 

The structure, findings and methodology are correctly presented. However, some concrete questions must be improved:

- state of the art about similar studies in rural areas must be improved: narrative of social change in rural areas, for example, authors like Vercher.

- sources of the tables do not mentioned below of these.

- it is necessary to be more precise on the methodology in a scheme, and the process followed. 

Author Response

We thank the reviewer for the opportunity to revise and resubmit our manuscript to Social Sciences. All the reviewers provided us with thoughtful and thorough evaluations of our paper. We believe that the revisions we have made based on these comments have allowed us to develop a much-improved manuscript. However, if we have not addressed specific comments, we are more than willing to make the necessary changes going forward. Our responses to the points raised are outlined below.

Reviewer 1:

Reviewer’s comment: The manuscript focuses on narratives about their community and community stories in Utah’s Heber Valley and social change. The structure, findings and methodology are correctly presented. Author’s response: Thank you for taking the time to carefully evaluate and summarize our manuscript.

Reviewer’s comment: Some concrete questions must be improved:

  • State of the art about similar studies in rural areas must be improved: narrative of social change in rural areas, for example, authors like Vercher [Nestor Vercher et al. (2021)]. Author’s response: The recommendation by the reviewer to include narratives about social change in rural areas was greatly appreciated.  Additional references that discuss the effect of social change in rural areas were added.
  •  Sources of the tables do not mentioned below of these. Authors’ response: The specifics of this request are unclear. However, we added the sources to each table to be thorough and are amenable to further additions.   
  •  It is necessary to be more precise on the methodology in a scheme, and the process followed.Authors’ response: we have made revisions throughout the methods sections to help clarify our approach.

Thank you for your candid comments. We have endeavored to the best of our ability to address all the concerns raised in the review. If we have misunderstood a comment or failed to sufficiently revise our manuscript, we are more than willing to make any necessary changes going forward.

Reviewer 2 Report

This is a methodologically rich, relevant and exciting paper examining the impacts of mega-events on community narratives. The methods are clear, extensive and appropriate for the research questions. Overall, the manuscript is well-written and could be beneficial to the field. I have a few suggestions for improvement:

·      My first suggestion would be to briefly mention the core findings in the abstract to provide context for readers.

·      The background section could benefit from a more detailed discussion on the notions of perceived community, the transition from one perceived community to another (what is precisely meant by this), and various experiences of place change. Also, it would be good to set the ground from the beginning regarding your epistemological approach to place and community, how you define them and how you frame their change.

·      The references are quite old; it would benefit the authors to consider more recent literature on community stories, community as narrative, and place change. A few suggestions that might be useful:

o   Devine‐Wright, P., 2009. Rethinking NIMBYism: The role of place attachment and place identity in explaining place‐protective action. Journal of community & applied social psychology19(6), pp.426-441.

o   Seamon, D., 2020. Place attachment and phenomenology: The dynamic complexity of place. In Place Attachment (pp. 29-44). Routledge.

o   Anton, C.E. and Lawrence, C., 2016. The relationship between place attachment, the theory of planned behaviour and residents’ response to place change. Journal of Environmental Psychology47, pp.145-154.

·      On the same note, the background section could be improved by situating the research in the wider context of studies on the social impact of mega-events and similar cases, e.g. London Olympics is a famous, extensively studied example.

·      The emphasis on the residents’ narrative and the intriguing idea of ‘community as story’ are somehow left underdeveloped in the rest of the paper. This can be expanded when discussing the findings from the textual analysis by highlighting these stories and narratives. As we read through the paper, the quantitative aspect of the survey analysis takes over while the rich textual data is sidelined.

·      A minor suggestion in terms of style: the authors could consider integrating direct quotes more seamlessly into the text, e.g. the quote from Cope et al. (2019) on page 2, lines 92 and 93.

·      The research objectives could be integrated into the introduction instead of giving them a separate section; it seems repetitive.

·      The N number on page 4, line 200, seems incorrect.

·      In section 5.2 the term ‘textural’ is used a couple of times instead of ‘textual’.

·      In terms of the survey data, it would be helpful if the authors could provide an appendix showing the questions included in the survey.

·      The textual data analysis could benefit from a more detailed discussion of the time factor. For instance, can you identify any change in the narratives before and after the event?  

·      Lines 520-521, aren’t the mentioned years about the survey data? It would be beneficial to also specify the publication years of the analysed newspapers in section 5.2.

·      Lastly, the discussion must be revised and better organised to better address the three RQs. It is hard to identify the authors’ responses to the question. The questions specifically ask about the 'transition' in the residents’ narratives, which could be better elaborated in the discussion.

Author Response

We thank the reviewer for the opportunity to revise and resubmit our manuscript to Social Sciences. All the reviewers provided us with thoughtful and thorough evaluations of our paper. We believe that the revisions we have made based on these comments have allowed us to develop a much-improved manuscript. However, if we have not addressed specific comments, we are more than willing to make the necessary changes going forward. Our responses to the points raised are outlined below.

Reviewer 2:

  1. Reviewer’s comment: This is a methodologically rich, relevant and exciting paper examining the impacts of mega events on community narratives. The methods are clear, extensive and appropriate for the research questions. Overall, the manuscript is well-written and could be beneficial to the field. Authors’ response: We greatly appreciate and recognize that work that went into this constructive review. Thank you.

  1. Reviewer’s comment: My first suggestion would be to briefly mention the core findings in the abstract to provide context for readers. Author’s response: Excellent point. A new sentence at the end of the abstract was added to address this.

  1. Reviewer’s comment: The background section could benefit from a more detailed discussion on the notions of perceived community, the transition from one perceived community to another (what is precisely meant by this), and various experiences of place change. Authors’ response: In the background section we added details to the discussion of perceived community, transitions from one community to another, and experiences of place change.

  1. Also, it would be good to set the ground from the beginning regarding your epistemological approach to place and community, how you define them and how you frame their change. Authors’ response: We agree that it is important to define community and place at the outset of the paper and added a paragraph at the beginning of the introduction to clarify the use of these terms.

  1. Reviewer’s comment: The references are quite old; it would benefit the authors to consider more recent literature on community stories, community as narrative, and place change. A few suggestions that might be useful: Devine-Wright (2009), Seamon (2020), and Anton and Lawrence (2016). Authors’ response: We appreciate these recommendations and have incorporated them in our manuscript where appropriate.

  1. Reviewer’s comment: On the same note, the background section could be improved by situating the research in the wider context of studies on the social impact of mega-event and similar cases, e.g. London Olympics is a famous, extensively studied example. Authors’ response: We added a paragraph to our background section to provide a wider context of studies on the social impacts of mega-events.

  1. Reviewer’s comment: The emphasis on the residents’ narrative and the intriguing idea of ‘community as story’ are somehow left underdeveloped in the rest of the paper. This can be expanded when discussing the findings from the textual analysis by highlighting these stories and narratives. Authors’ response: We agreed with this recommendation and added verbiage in the results section to highlight the continuity of this theme.

  1. Reviewer’s comment: As we read through the paper, the quantitative aspect of the survey analysis takes over while the rich textual data is sidelined. Authors’ response: This is a valid point. As this is a mixed methods article, it will inevitably have one aspect that outweighs another. Based on the data collected, we believe we have presented the results in the most objective manner. We are willing to make concrete changes should the reviewer provide additional recommendations for how this can be accomplished.

  1. Reviewer’s comment: A minor suggestion in terms of style: the authors could consider integrating direct quotes more seamlessly into the text, e.g. the quote from Cope et al. (2019) on page 2, lines 92 and 93. Authors’ response: The quotes are now integrated more seamlessly into the text.

  1. Reviewer’s comment: The research objectives could be integrated into the introduction instead of giving them a separate section; it seems repetitive. Authors’ response: We agree with the reviewer, and we have integrated the research objectives into the introduction. We have deleted the Research Objectives section and re-numbered accordingly.

  1. Reviewer’s comment: The N number on page 4, line 200, seems incorrect. Authors’ response: Thank you for the keen eye for detail. We have corrected this to N=3,285.

  1. Reviewer’s comment: In section 5.2 the term ‘textural’ is used a couple of times instead of ‘textual’. Authors’ response: Thank you for your attention to detail. We have corrected the spelling of these terms and ensured all other uses of textual are correct.

  1. Reviewer’s comment: In terms of the survey data, it would be helpful if the authors could provide an appendix showing the questions included in the survey. Authors’ response: Due to the large range of data collection from multiple waves, over about 30 years, and conducted by multiple researchers, an appendix with all of the questions is not feasible. We can provide a footnote that says information is available upon request.

  1. Reviewer’s comment: The textual data analysis could benefit from a more detailed discussion of the time factor. For instance, can you identify any change in the narratives before and after the event? Authors’ response: This is an important element of our paper--change over time. Additional text and many revisions were made throughout the paper, especially in the results section, to elucidate change over time.

  1. Reviewer’s comment: Lines 520-521, aren’t the mentioned years about the survey data? It would be beneficial to also specify the publication years of the analysed newspapers in section 5.2. Authors’ response: Thank you for this recommendation. Because we would need to include the prior year and year of the HVCS, it was determined that adding the years of the survey data would suffice.
  2. Reviewer’s comment: Lastly, the discussion must be revised and better organized to better address the three RQs. It is hard to identify the authors’ responses to the question. The questions specifically ask about the ‘transition’ in the residents’ narratives, which could be better elaborated in the discussion. Authors’ response: We rewrote and reorganized our discussion section to better address our three research questions and to discuss the “transition” in the residents’ narratives.

Thank you for your candid comments. We have endeavored to the best of our ability to address all the concerns raised in the review. If we have misunderstood a comment or failed to sufficiently revise our communication manuscript, we are more than willing to make any necessary changes going forward.